# Infection patterns in simple and complex contagion processes on networks

**Diego Andrés Contreras**[1⊙], **Giulia Cencetti**[1,2⊙]*, **Alain Barrat**[1]

**1** Aix-Marseille Univ, Université de Toulon, CNRS, Centre de Physique Théorique, Turing Center for Living Systems, Marseille, France, **2** Fondazione Bruno Kessler, Trento, Italy

⊙ These authors contributed equally to this work.
* giulia.cencetti@cpt.univ-mrs.fr

**Data Availability Statement:** All data are available at: http://www.sociopatterns.org. The code used for numerical simulations and analyses of infection

## Abstract

Contagion processes, representing the spread of infectious diseases, information, or social behaviors, are often schematized as taking place on networks, which encode for instance the interactions between individuals. The impact of the network structure on spreading process has been widely investigated, but not the reverse question: do different processes unfolding on a given network lead to different infection patterns? How do the infection patterns depend on a model's parameters or on the nature of the contagion processes? Here we address this issue by investigating the infection patterns for a variety of models. In simple contagion processes, where contagion events involve one connection at a time, we find that the infection patterns are extremely robust across models and parameters. In complex contagion models instead, in which multiple interactions are needed for a contagion event, non-trivial dependencies on models parameters emerge, as the infection pattern depends on the interplay between pairwise and group contagions. In models involving threshold mechanisms moreover, slight parameter changes can significantly impact the spreading paths. Our results show that it is possible to study crucial features of a spread from schematized models, and inform us on the variations between spreading patterns in processes of different nature.

## Author summary

Contagion processes, representing the spread of infectious diseases, information, or social behaviors, are often schematized as taking place on networks, which encode for instance the interactions between individuals. We here observe how the network is explored by the contagion process, i.e. which links are used for contagions and how frequently. The resulting infection pattern depends on the chosen infection model but surprisingly not all the parameters and models features play a role in the infection pattern. We discover for instance that in simple contagion processes, where contagion events involve one connection at a time, the infection patterns are extremely robust across models and parameters. This has consequences in the role of models in decision-making, as it implies that numerical simulations of simple contagion processes using simplified settings can bring important insights even in the case of a new emerging disease whose properties are not yet well

patterns is publicly available at: https://github.com/giuliacencetti/Infection_pattern.

**Funding:** This work was supported by the Agence Nationale de la Recherche (ANR) project DATAREDUX (ANR-19-CE46-0008). G.C. acknowledges the support of the European Union's Horizon 2020 research and innovation program under the Marie Skłodowska-Curie grant agreement No 101103026. The funders had no role in study design, data collection and analysis, decision to publish, or preparation of the manuscript.

**Competing interests:** The authors have declared that no competing interests exist.

known. In complex contagion models instead, in which multiple interactions are needed for a contagion event, non-trivial dependencies on model parameters emerge and infection patterns cannot be confused with those observed for simple contagion.

## Introduction

Contagion processes pervade our societies. Examples include the spread of infectious diseases, both through contacts between hosts and following their mobility patterns, but also information diffusion or the propagation of social behavior [1–6]. Modeling of these processes often includes a description of the interactions among the hosts as a network, in which nodes represent individuals and a link between nodes correspond to the existence of an interaction along which the disease (or information) can spread. In the resulting field of network epidemiology [4, 6, 7], many results have been obtained for the paradigmatic models of diffusion processes, in which the hosts can only be in a few possible states or compartments, such as susceptible (S, healthy), infectious (I, having the disease/information and able to transmit it), or recovered (R, cured and immunized) [1, 2]. These results concern mainly the context of models aimed at describing the spread of infectious diseases, represented as so-called *simple contagion* processes: namely, processes in which a single interaction between a susceptible and an infectious can lead to a transmission event [1, 6]. In this context, many studies have provided insights into how the structure of the underlying network influences the spread and impacts the epidemic threshold (separating a phase in which the epidemic dies out from one in which it impacts a relevant fraction of the population), and how various containment strategies can mitigate the spread [4, 6].

Fewer results concern the detailed analysis of the process dynamics and spreading patterns, despite its relevance [8]. In particular, the reverse question of whether different processes lead to different or similar infection patterns has barely been explored. At the population level, a robustness of the shapes of the epidemic curves has been observed for various spreading models [9, 10] and contact networks [11]. In heterogeneous networks, it has also been shown that simple contagion spreading processes first reach nodes with many neighbours, and then cascade towards nodes of smaller degree [12–14]. Moreover, in the context of metapopulation models, in which each node of the network represents a geographic area and hosts can travel between nodes on the network, possibly propagating a disease, the heterogeneity of travel patterns has been shown to determine dominant paths of possible propagation at the worldwide level [8, 15, 16], allowing for instance to provide predictions for the arrival time of a pandemic in various parts of the world [17, 18].

In addition, while these results concern simple contagion processes, it is now well known that such models might not be adequate to describe some contagion mechanisms, such as social contagion of behaviors. Empirical evidence has led to the definition and study of models of *complex contagion* [3, 19]: in these models, each transmission event requires interactions with multiple infectious hosts. In particular, models involving threshold phenomena [20] or group (higher-order) interactions [21] have been put forward, but results concerning the detail of their propagation patterns are scarce [14, 22].

Overall, most results on propagation patterns concern simplified models with few compartments (such as the susceptible-infected-susceptible (SIS) and susceptible-infected-recovered (SIR)) and simple contagion processes. The question arises thus of their applicability to more realistic models and to other types of spreading processes, and of the possibility to directly apply them in concrete cases. Here we contribute to tackle these issues by investigating

spreading patterns for different types of contagion models on networks and hypergraphs and by addressing the following questions: how general are the propagation patterns observed in these models, and are they similar in more realistic models with compartments including latent individuals, asymptomatic cases, etc? How well do propagation patterns of simple contagion inform us on complex contagion ones, and do the most important seeds or the nodes most easily reached differ depending on the precise model or type of contagion?

To this aim, we consider the infection network of a process [8], which gives the probability of a node to be directly infected by another one, averaged over realizations of the process, and generalize it as well to complex contagion models. We compare the resulting patterns within each model as its parameters change, between different models of simple contagion and between different types of contagion processes. We first find an extreme robustness of the contagion patterns across models of simple contagion. These patterns slightly depend on the reproductive number of the spread, but are almost completely determined by the final epidemic size. This indicates also that one can define spreader and receiver indices to quantify a node's tendency to contaminate or be contaminated by its neighbours: these indices are largely independent of the specific disease model and can thus be computed on simple cases with arbitrary parameters. The situation changes when models of complex contagion are considered. On the one hand, patterns of contagion turn out to be less robust in threshold models. On the other hand, they depend on the interplay between pairwise and group processes for models involving higher-order interactions.

## Results

### General framework

We consider the context of network epidemiology, i.e., of spreading processes on a weighted network where nodes represent the hosts and weighted links between the hosts correspond to contacts along which a disease can spread, with probability depending on the link weight [6]. Specifically, the weighted networks we will use to perform numerical simulations of spreading processes are empirical networks obtained by temporally aggregating time-resolved data describing contacts between individuals in various contexts [23–25], where the weight $W_{ij}$ between two individuals $i$ and $j$ is given by their total interaction time (see Methods).

On these networks, we will first consider several models of simple contagion, in which each node can be in several states such as susceptible, latent, infectious, and recovered, and an infectious node can transmit the infection to a susceptible neighbour with a certain probability per unit time. We will consider models with different sets of states, corresponding both to very schematic and to more realistic situations, and both Markovian and non-Markovian processes. On the other hand, we will consider a model of complex contagion that involves higher-order contagion mechanisms, i.e., interactions among groups of nodes [21]: This model describes the fact that the probability of a contagion event can be reinforced by group effects, and is defined on hypergraphs [26] in which interactions can occur not only in pairs but also in larger groups. It has indeed been shown that the inclusion of such effects leads to an important phenomenological change, with the emergence of a discontinuous epidemic transition and of critical mass phenomena. Finally, we will also consider so-called threshold models [20], in which a susceptible node becomes infected when the fraction of its interactions spent with infected neighbors reaches a threshold $\theta$, to mimic the fact that an individual may adopt an innovation only if enough friends are already adopters. All models and their parameters are described in detail in the Methods section, and their mechanisms are sketched in Fig 1.

For each given spreading model and propagation substrate (network or hypergraph), we perform numerical (Monte Carlo) simulations of the spread at given parameter values, starting

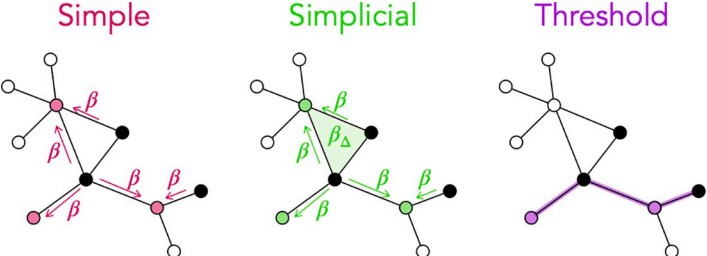

**Fig 1. Sketch of the models of contagion considered.** In all sketches, black nodes represent infectious hosts, empty nodes are susceptible, and colored nodes represent the hosts that can be contaminated by the infectious ones. Left: Simple contagion on weighted graphs. Contagion events occur along the network edges, with probability per unit time given by $\beta$ multiplied by the weight $W_{ij}$ of the edge $(i, j)$ between a susceptible and an infected node. Center: Simplicial model on weighted hypergraphs. Contagions can take place both along network edges (rate $\beta W_{ij}$) and if a susceptible node $i$ is part of a group $(i, j, k)$ with $j$ and $k$ both infectious (rate $\beta_\Delta W_{ijk}^\Delta$, with $W_{ijk}^\Delta$ the weight of the hyperedge $(i, j, k)$). Right: Threshold model on weighted graphs. A susceptible node becomes infected when the sum of the weights of its connections with infected nodes, divided by the total weight of its connections, exceeds a threshold $\theta$.

from a single infectious seed taken at random in the network, while all other nodes are susceptible (see Methods). The *infection pattern* of the model is then the weighted and directed graph **C** such that $C_{ij}$ is the probability (averaged over 1000 realizations of the spread) that node $i$ infected node $j$ [8, 16]. In practice, it is obtained from the numerical simulations, by counting all the direct infectious events from $i$ towards $j$ among all runs, and dividing by the number of runs. The infection pattern hence represents the signature of an epidemic, highlighting the paths that are taken by the contagion process with a higher probability. **C** was defined for metapopulation models [8, 16] as the probability for a contagion to arrive in a geographical area from another one. Here we consider the case instead in which nodes represent hosts; moreover, this definition needs to be generalized in the case of complex contagion processes where the contagion of a node originates from several others, as described later. We first note that a non-zero $C_{ij} > 0$ can be obtained if and only if there exists an interaction between $i$ and $j$ in the weighted network; moreover, one can expect that the probability $C_{ij}$ of $i$ infecting $j$ depends on the weight $W_{ij}$ of their connection. However, it also depends on the probability of $i$ to be infected in the first place, to be infected before $j$, and of $j$ not to be infected through another interaction. Overall, one can thus expect $C_{ij}$ to depend on non trivial properties of the network topology and not only on the weight of the link between $i$ and $j$. In particular, even if the interaction weights are symmetric, this is not a priori the case for the infection pattern: the network defined by the matrix $C_{ij}$ is directed. This is shown in Fig 2 for a toy network, where the largest values of $C_{ij}$ do not correspond to the largest link weights. Once **C** is defined, we can moreover use it to compute spreader and receiver indices for each node, respectively as $s_i = \sum_j C_{ij}$ and $r_i = \sum_j C_{ji}$, i.e., as the out-strength and in-strength of each node in the directed network of the infection pattern.

It is worth noting here that **C**, and as a consequence also the spreader and receiver indices, depend both on the specific model of spread and on its parameters. We will explore these dependencies in detail in the following sections. In this exploration, we have considered, as the support of the contagion models we investigate, data describing contacts between individuals collected in a conference [27], a hospital [28], a workplace [27], a primary school [29] and a high school [30]. The primary school contact data has been used in various studies to feed numerical simulations of infectious diseases' models [31–33] and entails rich intertwined structural and temporal features such as groups of temporarily densely connected nodes and alternating patterns of nodes being structured in groups or able to connect in a more global

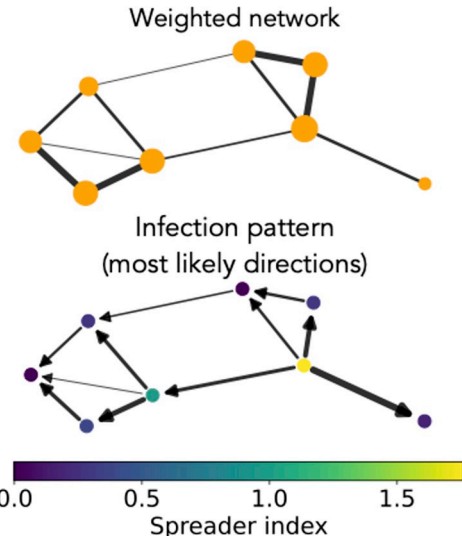

**Fig 2. Simple contagion.** Toy network illustrating the asymmetry of the infection pattern and its dissimilarity with the adjacency matrix. The upper sketch shows the weighted adjacency matrix (links' width proportional to their weights, nodes' size proportional to their weighted degree). The lower sketch represents the infection pattern for a simple SIR contagion with $R_0 = 2$ (averaged over 500 simulations). For each connection only the direction with higher probability of infection is shown and the arrows' width is proportional to the probability. The nodes are colored according to their spreader index.

manner, as well as an important number of simultaneous group (higher-order) interactions [29, 34–37]. We thus show in the main text the results obtained for this data set, and we show the results for the other data sets in the S1 Text.

## Simple contagion

We consider several models of simple contagion, characterized by different sets of possible states for the hosts and various types of dynamics between states. The simplest is the Susceptible-Infected-Recovered (SIR) model, in which a susceptible individual $i$ (S) can become infected (I) with rate $\beta W_{ij}$ when linked with another infected individual $j$ by an edge of weight $W_{ij}$ (see Fig 1). Infected individuals then spontaneously become recovered (R) with rate $\mu_I$ and cannot participate in the dynamics anymore. The most studied extension of this model is the SEIR one, in which susceptible individuals become exposed (E, not yet contagious) with rate $\beta$ upon contact with an I individual, before becoming infected. In both SIR and SEIR, we consider on the one hand fixed rates of transition from the I to the R state and from the E to the I state; the times that an individual spends in the E and I states, resp. $\tau_E$ and $\tau_I$, are then exponentially distributed random variables (with averages given by the inverses of the transition rates). A more realistic dynamical process is obtained by a non-Markovian dynamics between these states, in which $\tau_E$ and $\tau_I$ are random variables taken from Gamma distributions with given mean and standard deviation. As both SIR and SEIR remain generic models, we also consider a more elaborate model designed to represent the propagation of COVID-19, in which individuals can be exposed and not contagious, pre-symptomatic but already infectious, infectious but asymptomatic, or infectious and symptomatic [13, 33]. These models and their parameters are described in more detail in the Methods section.

For each model and network, once the parameters of the spontaneous transitions are fixed, it is possible to adjust the contagion rate $\beta$ to obtain a specific value of the reproductive

number $R_0$, defined as the expected number of cases directly generated by one initial infected individual in a population where all other individuals are susceptible to infection [1]. For each model and parameter value, we compute the infection pattern $C$ and the spreader and receiver indices of each node as explained above.

As expected and anticipated in the toy example of Fig 2, we find that the matrix $\mathbf{C}$ is asymmetric, and we show the similarity of its elements with the weighted adjacency matrix of the underlying network in the Section A of S1 Text. We then compare in Fig 3A the infection patterns $\mathbf{C}$ obtained in different simple contagion models, calibrated so as to correspond to the same value of $R_0$. The comparison is performed by computing the cosine similarity between the lists of elements of the matrices $\mathbf{C}$ obtained in the various cases (see Methods for the definition of cosine similarity). Even at fixed $R_0$, each model entails a different time evolution of the epidemic (see Section B in S1 Text) with a different spreading velocity, and also different compartments, so corresponds to a different general process. One could hence suppose that the infection pattern could also be largely different from one model to the next. However, Fig 3A highlights how the infection patterns are actually extremely similar across models, with similarity values above 0.98. Hence, the probability for each network link of being used for a contagion event is largely independent of the specific contagion model considered (at given $R_0$), despite the differences in their temporal evolution. In other words, contagion paths are not only stable within one model [16] but also across models. In the following analysis, we will thus focus on the simplest SIR model.

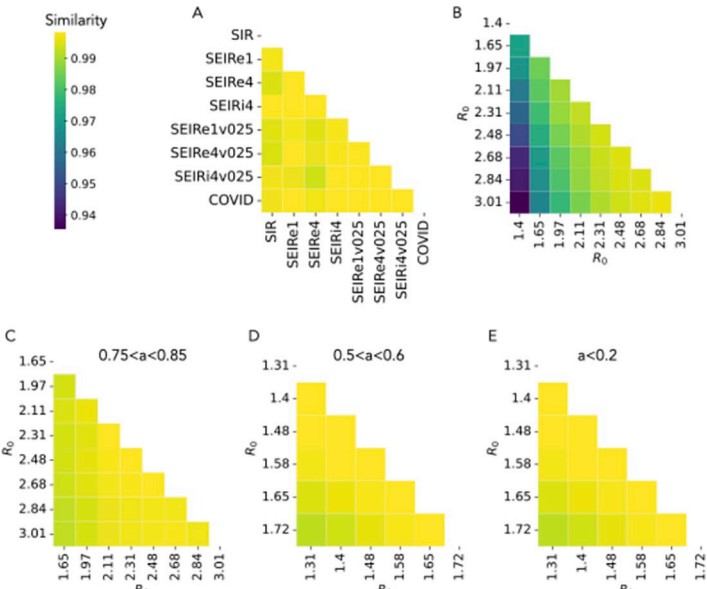

**Fig 3. Simple contagion.** A: Cosine similarity between the infection patterns of different models of simple contagion, simulated with the same $R_0 = 2.5$ (see Methods for the description of the models). B: Cosine similarity between the infection patterns obtained at varying $R_0$ for the SIR model of simple contagion. C: Cosine similarity between infection patterns at varying $R_0$ for the SIR model, with infection patterns computed only using runs with final attack rate between 0.75 and 0.85. D: Same as C but using runs with final attack rate between 0.5 and 0.6. E: Same as C but using runs with final attack rate lower than 0.2. The results in panel C have been obtained by comparing, for each value of $R_0$, infection patterns obtained by averaging over 1000 simulations with final attack rate $a$ in the chosen range. For panels D and E the number of simulations to average on has been increased to 10000 and 50000, respectively. Indeed, smaller values of $a$ mean that less nodes and links are involved in each run, so that one needs to average over more runs to compute the infection probability for each link.

Fig 3B reports the cosine similarity between matrices **C** obtained with the SIR model at varying $R_0$. Interestingly, although the similarity values are very large, they are lower than between models at fixed $R_0$, revealing a weak dependency of the infection patterns on $R_0$. To understand this point further, it is worth reminding that, while $R_0$ largely determines the initial velocity of the spread, the contagion process remains stochastic, and simulations with a fixed $R_0$ can lead to different final attack rates, i.e., final values of the density of recovered individuals once the spreading process is over, i.e., once no contagion can take place any longer (we show in the Section E of S1 Text the resulting distributions of final attack rates for several values of $R_0$). We thus consider the infection patterns at different values of $R_0$ but at fixed final attack rate. To this aim, we need to consider compatible ranges of $R_0$ and final attack rates, i.e., a range of attack rates that can be reached at all the values of $R_0$ used. We report in Fig 3C the analogous of Fig 3B, but where the matrices **C** have been computed taking into account only the simulations with a final attack rate between 0.75 and 0.85 (as shown in the S1 Text, such final attack rates are reached by a non-negligible fraction of the runs for $R_0$ between 1.65 and 3). The similarity values become larger than 0.99, suggesting that the infection pattern of a spreading model mostly depends on its average final attack rate. To check the generality of this result, we extend this investigation to two other ranges of final attack rates in Fig 3D and 3E, namely $0.5 - 0.6$ and $0 - 0.2$ (note that, to obtain enough simulations with final attack rates between 0.5 and 0.6, we need to consider lower values of $R_0$). We obtain also in theses cases very high values of the similarity.

Such results moreover lead us to an additional investigation, based on two simple points: (i) the final attack rate is an increasing function of $R_0$ and (ii) for a given $R_0$, the average attack rate is a continuously increasing function of time, which thus passes through the values of the final attack rates obtained with lower values of $R_0$. The question arising is thus the following: if we consider, for a large $R_0$, the time-dependent infection patterns **C**$(t)$ (obtained by averaging on all infection events up to $t$), are the matrices **C**$(t)$ similar to the final infection patterns obtained with lower values of $R_0$?

We investigate this issue in Fig 4 through the following procedure. First, we consider as reference an SIR model with $R_0^{ref} = 4$, and perform 1000 simulations of this model. At each time, we build the time-dependent **C**$_{ref}(t)$ by averaging on all the contagion events occurred in these 1000 simulations up to $t$. Second, we consider several lower values of $R_0$, namely $R_0 \in \{1.5, 2, 2.5, 3, 3.5\}$, perform 1000 simulations for each value, and compute the resulting infection patterns **C**$_{R_0}$. The top panel of Fig 4 displays the similarity between the time-dependent infection pattern for $R_0 = 4$, **C**$_{ref}(t)$, and the final infection patterns obtained with the lower values of $R_0$, **C**$_{R_0}$. Each such similarity goes through a maximum (with large values above 0.98) as a function of time, and this maximum is obtained when the time-dependent attack rate of the reference process ($R_0 = 4$) is almost equal to the final attack rate of the process at lower $R_0$, as seen in the middle panel of Fig 4.

More precisely, the 1000 simulations of the reference model yield a distribution of attack rates at each time $t$ (displayed in black in the bottom panels of Fig 4 for five different times). These distributions are typically bimodal and the location of the non-zero mode for each time is plotted in the middle panel of Fig 4 (black curve). The colored dots correspond instead to the non-zero modes of the distributions of final attack rates for the lower $R_0$ values (full distributions shown by the colored curves in the bottom panels, obtained as well with 1000 simulations of the model for each $R_0$). The y-value of each coloured dot is reached by the black curve in the middle panel at the same time as the maximum of the corresponding similarity curve in the top panel. Note that the fact that the similarity between **C**$_{ref}(t)$ and **C**$_{R_0}$ does not reach 1

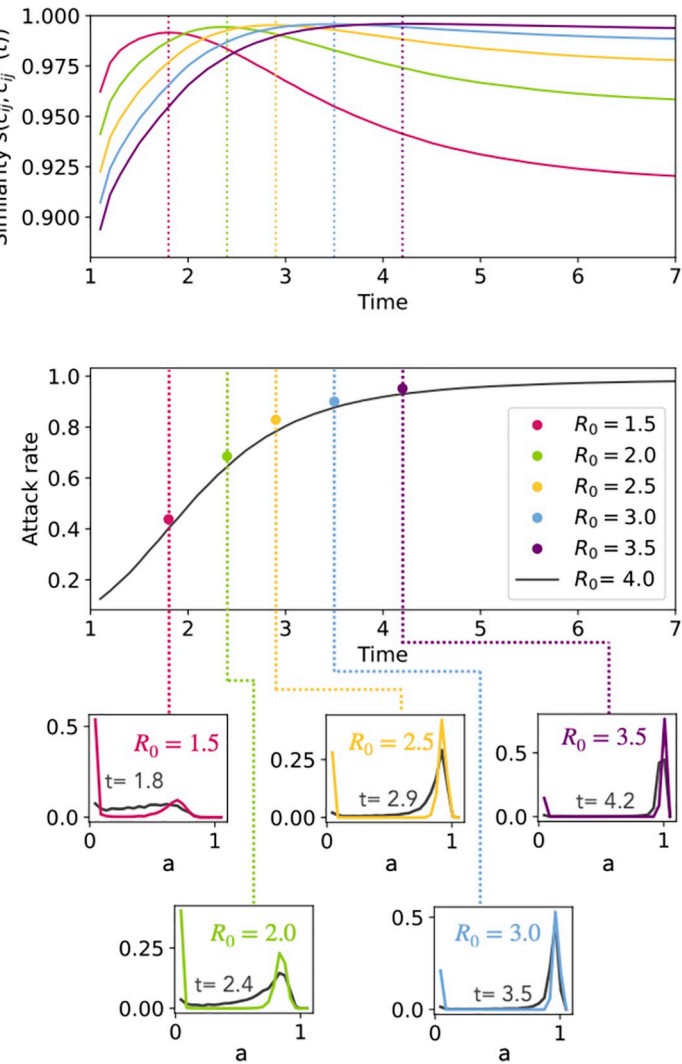

**Fig 4. Simple contagion.** Comparison, for the SIR model, between a reference $R_0 = 4$ and five testing parameter values ($R_0$ from 1.5 to 3.5). Each curve in the upper panel represents the similarity in time between the temporal infection pattern $\mathbf{C}_{ref}(t)$ of the reference and the infection pattern $\mathbf{C}_{R_0}$ of each testing parameter. $\mathbf{C}_{ref}(t)$ is computed by averaging, over 1000 numerical simulations of the SIR model at $R_0 = 4$, the contagion events occurring until $t$. $\mathbf{C}_{R_0}$ is instead obtained by averaging all contagion events of 1000 numerical simulations of the SIR model at $R_0$. The middle panel shows as a black curve the temporal evolution of the non-zero mode of the distributions of attack rates of the reference spread, also computed over all 1000 simulations at $R_0 = 4$ and at each time. The colored dots show, for each $R_0 \in \{1.5, 2, 2.5, 3, 3.5\}$, the value of the non-zero mode of the final attack rate distribution, computed over 1000 simulations at each $R_0$. The corresponding attack rate distributions are shown in the smaller panels below.

can be explained by the fact that the distributions of time-dependent and final attack rates do not coincide completely.

In other words, at each time step $t$ of a contagion process with a high $R_0$, the partial infection patterns, which describe the contagion probability of each connection until $t$, are extremely similar to the full infection patterns of a process with a lower value of $R_0$. Vice-versa, this also means that the infection patterns of processes with low $R_0$ can be approximated extremely well by using a single process at large $R_0$ and computing its time-dependent infection patterns.

We finally also show in the S1 Text that the range of values of the spreader and receiver indices depend on the reproductive number $R_0$, but the ranking of nodes by these indices is very robust across models and across values of $R_0$. Moreover, when fixing the attack rate, the ranges of values become equivalent even for different $R_0$, and the ranking of nodes becomes almost independent of $R_0$, showing that also this ranking is almost completely determined by the attack rate, and in any case very robust across parameter values. Overall, our results indicate an extreme robustness of the infection patterns across different models of simple contagion, despite their diversity in the sets of possible states for the hosts and of dynamical transition rules. Moreover, while the infection pattern does depend (very) slightly on the model parameters, it is almost completely determined by the final attack rate of the process. This result is not valid for complex contagion processes, as we will see in the next sections.

## Simplicial contagion

Let us now consider a model of complex contagion in which the propagation can occur both on the links of the network, as in the case of simple contagion, but also on higher order (group) interactions, namely the simplicial contagion model [21], generalized here to weighted hypergraphs. As in [21], we limit ourselves for simplicity to contagion processes on first and second order interactions (pairs and triads), neglecting structures of higher orders, which will only appear as decomposed into links and triangles. We consider a SIR model, where a susceptible host $i$ can receive the infection (i) with rate $\beta_| W_{ij}$ when sharing a link of weight $W_{ij}$ with an infected host $j$, and (ii) with rate $\beta_\Delta W_{ikl}^\Delta$ when part of a group $i$, $k$, $l$ of three interacting nodes such that both $k$ and $l$ are infected ($W_{ikl}^\Delta$ being the weight of the hyperedge ($i$, $k$, $l$), see Methods and Fig 1). As in simple contagion models, infected nodes recover spontaneously—we consider here a fixed recovery rate $\mu_I$.

As contagion events can occur both through links and triads, we here need to generalize the computation of **C** by defining the number of infection events from $i$ to $j$, $n_{i \to j}$, as follows: if $j$ is infected by $i$ in a pairwise interaction, $n_{i \to j}$ is incremented by one; if instead $j$ is infected through a triadic interaction with $i$ and $l$ who are both infected, $i$ and $l$ play an equivalent role in this contagion event, and thus we divide the "responsibility" of the event equally among them: both $n_{i \to j}$ and $n_{l \to j}$ are incremented by 1/2. $C_{ij}$ is finally the ratio of $n_{i \to j}$ to the number of numerical simulations considered.

While there is a one-to-one correspondence between $R_0$ and the infection rate $\beta$ in the case of simple contagion (the other parameters being fixed), a given $R_0$ could here correspond to various pairs ($\beta_|$, $\beta_\Delta$). We thus compare the infection patterns obtained when varying both parameters in Fig 5A, going from a situation in which the contagion events occur mostly on triads to one in which they occur mostly on links (as shown in Fig 5B). These different ratios between the two parameters $\beta_|$ and $\beta_\Delta$, given they yield different relative abundances of the two types of infection (simple vs complex), can be expected to give rise to different infection patterns. The similarity values obtained remain however high, even between the most extreme cases (very different relative values of the numbers of infections in pairs and triads). We show in the S1 Text results concerning the receiver and spreader indices and the subsequent ranking of nodes: similarly to the case of simple contagion, the ranking of nodes are very robust across parameter values, even if the range of values taken by the indices change. This can be explained by the observation that, in social networks, higher order interactions and pairwise ones largely overlap, i.e., nodes connected in groups with large weights are typically also connected by links with large weights (see Section C in S1 Text). The infection patterns on pairwise links and on triads thus also overlap. In fact, the similarity between the infection patterns of the simple SIR contagion process and the simplicial one, shown in Fig 5E at varying $R_0$ (of the simple

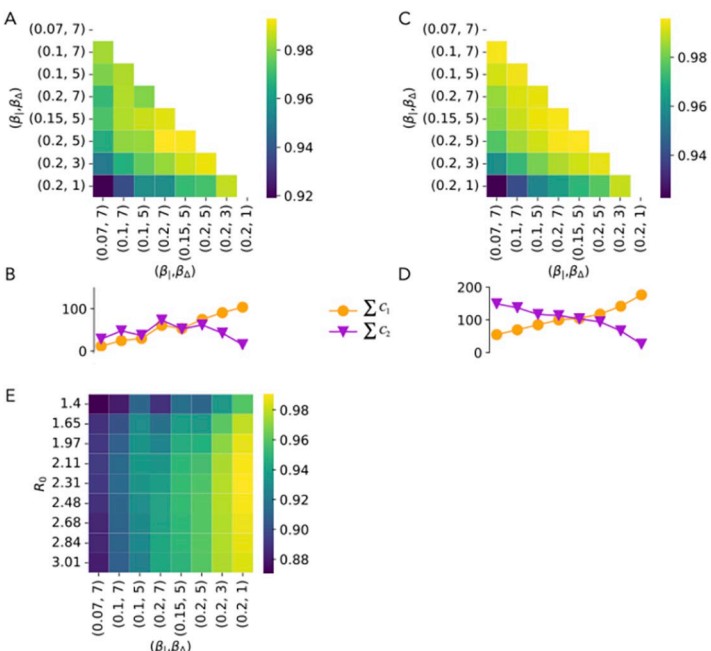

**Fig 5. Simplicial contagion.** A: Cosine similarity between infection patterns at varying different combinations of $\beta_|$ and $\beta_\Delta$. B: Number of contagions taking place via first and second order simplices in the simulations of the previous panel. $\mathbf{C}_1$ is the infection pattern matrix obtained considering only infections taking place via pairwise links and $\mathbf{C}_2$ is the analogous for triads infections, with $\mathbf{C}_1 + \mathbf{C}_2 = \mathbf{C}$. In the plot we report the sum of all elements of the matrices $\Sigma_{ij}(\mathbf{C}_1)_{ij}$ and $\Sigma_{ij}(\mathbf{C}_2)_{ij}$, which give the respective fractions of contagion events of each type. C: Cosine similarity between infection patterns at varying different combinations of $\beta$ and $\beta_\Delta$, when computing the infection patterns using only simulations with attack rate between 0.6 and 0.7. D: Number of contagions taking place via first and second order simplices in the simulations of the previous panel. E: Cosine similarity between infection patterns of simplicial contagion (for the same range of values of $\beta_|$ and $\beta_\Delta$) and simple contagion (for different values of $R_0$).

contagion) and parameters ($\beta_|$, $\beta_\Delta$), are also high, especially when the pairwise contagion events dominate in the simplicial model.

An interesting distinction with the case of simple contagion is however revealed in Fig 5. Namely, while the infection pattern of a simple contagion process is almost completely determined when fixing its final attack rate (see Fig 3), this is not the case for the simplicial one. We show indeed in Fig 5C the similarity between infection patterns at different values of the spreading rates, but when these patterns are computed using only simulations with a given final attack rate. In contrast to the case of simple contagion, constraining the attack rate does not change the similarity values, which remain similar to the ones observed in Fig 5A. This is clearly due to the fact that the same attack rate can be obtained through very different relative numbers of pairwise and higher order infection events (Fig 5D). The differences between simplicial contagion infection patterns at different parameters measured in Fig 5A are thus mostly due to the differences in the combination between the two competing processes at work in this model (first-order vs. second-order contagions).

The simple and simplicial models entail fundamentally different contagion mechanisms, leading to different physics and different types of phase transitions, including critical mass phenomena [21, 26]. Here indeed, the differences in infection patterns are driven by the differences between pairwise and higher order contagions. However, the resulting infection patterns remain very similar in our simulations, which is probably largely due to the fact that, in the empirical data we consider, links and higher order hyperedges largely overlap, with correlated

weights (see S1 Text and [38]) so that both simple and higher order mechanisms tend to use the same infection routes. We confirm this hypothesis in the S1 Text by showing that, if correlations between the weights of links and higher order hyperedges are removed, the similarity between the infection patterns of simple and simplicial contagion notably decreases.

## Threshold contagion

We finally investigate the infection patterns resulting from a model of complex contagion driven by threshold effects on a network: in this model [20], a susceptible node can become infected (deterministically) only if the fraction of its neighbors that are infected overcomes a certain threshold $\theta$, the parameter of the process (see Fig 1). In the generalization of this model to weighted networks, a susceptible node becomes infected when the weight of its connections with infected nodes divided by the total weight of its connections exceeds the threshold. We moreover introduce a recovery parameter $\mu_I$ as in the previous cases, in order to obtain an SIR model as well. As in the simplicial model, the infection of a node $i$ is typically due to more than one other node. We thus generalize the computation of the infection pattern **C** similarly to the previous case: if $i$ becomes infected because $k$ of its neighbours $i_1, i_2, \ldots i_k$ are infected, each $C_{i_a i}$ is incremented by $W_{i_a i}/\sum_{b=1}^{k} W_{i_b i}$, i.e., by the relative contribution of $i_a$ to the infection event.

We compare the infection patterns of this model at various values of the parameter $\theta$ in Fig 6A. Interestingly, the values of the cosine similarity between patterns are still high, but typically much lower than in the previous cases, suggesting that the parameter $\theta$ plays a stronger role in determining the infection pattern than $\beta$ (or $R_0$) in simple contagion processes (see S1 Text for results on the receiver and spreader indices). This can be understood by the following argument: in simple contagion, all existing paths on the network can potentially support a contagion; on the other hand, changing the value of $\theta$ corresponds to allowing some infection patterns and impeding others, as it can change the number of infected neighbors needed to infect a given node. Smaller values of $\theta$ imply an easier and faster infection of nodes, while larger values only allow contagion of nodes connected with many infected, thus constraining infection to follow more specific patterns.

In Fig 6B we also compare the infection patterns of the threshold contagion model with the ones of simple contagion, showing that the two processes are characterized by rather different infection patterns. The similarity is higher for larger values of $\theta$: as $\theta$ becomes large, the condition needed for the infection of a node $i$ becomes stricter and can be fulfilled only if the neighbours $j$ to which $i$ is linked by its largest weights are infected. Thus, the infection pattern becomes closer to the one of a simple process.

In general, the infection patterns for the threshold model show a higher parameter dependency with respect to the simple models. However, the values of the similarities between

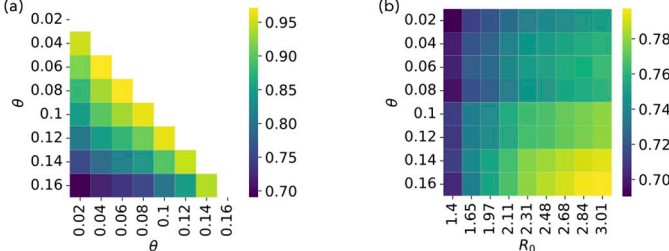

**Fig 6. Threshold contagion.** A: Cosine similarity between infection patterns at varying $\theta$. B: Cosine similarity between infection patterns of threshold contagion (for different values of $\theta$) and simple contagion (for different values of $R_0$).

infection patterns obtained in Fig 6 remain rather high, typically above 0.7. This is due to the fact that in all cases the infection patterns largely depend on (and are correlated with) the underlying weighted adjacency matrix (see Section A in S1 Text).

## Discussion

We have here investigated the infection patterns of various models of contagion processes on networks, using as substrate several empirical networks of contacts between individuals. In particular, while it is well known that the network structure impacts the spreading patterns, the question of how these patterns depend on the type of model considered (e.g., schematic or more realistic set of compartments, Markovian dynamics or more realistic transitions), on a model's parameters, or on the type of spreading process considered (i.e., simple vs. complex contagion) has been much less considered. Understanding these issues has however important consequences in the articulation between modeling and decision making, as modeling and theoretical investigations often focus on simple models with arbitrary parameters, while one could argue that decision making should be based on models as realistic as possible. Here, we have shed light on these questions by investigating the infection patterns, defined as measuring for each connected pair of nodes of the network the probability that an infection event occurs from one to the other [8, 16].

We have obtained results along four main directions. First, we have shown that these patterns are extremely robust in models of simple contagion. This robustness is in agreement with previous results obtained each within one specific model, such as the existence of a pattern of cascading dynamics from hubs towards less connected nodes in paradigmatic models of spread [12, 13, 39], or the possibility to define epidemic pathways making the spreading pattern of a disease on a network quite predictable [8, 15, 16]. These results also rationalize the fact that arrival times of a disease spread on a network can be obtained from purely topological measures [17]. We here extend however significantly previous literature by generalizing the robustness across a large ensemble of possible models typically used to describe the evolution of infectious diseases, even if they differ in the compartments used, in the parameters and, as a result, in the resulting dynamics timescales. In particular, within one model the spreading patterns slightly depend on the reproductive number but are almost fully determined by the final attack rate.

Second, the infection patterns also allow us to define a receiver and a spreader indices for each node, which give a ranking of nodes according to their relative risk of becoming infected during the spread and to spread to other nodes. The corresponding ranking of nodes is also very robust across models and parameters. Interestingly, this result gives support to, and puts on a firmer ground, a wealth of previous literature using topological centrality measures to predict epidemic sizes or to determine which nodes would be the best "sentinels" (i.e., nodes easily reached by a disease and hence to monitor more closely in a surveillance program). Most such studies indeed use very simple spreading models with often arbitrary parameters [40–44], and our results explain why correlations between a topological centrality and measures of epidemic impact are robust against parameter changes [45], making it indeed possible to limit such studies to a restricted set of models and parameters.

Third, we have generalized the infection patterns to complex contagion processes (typically used to describe social contagion) in which each contagion event can involve several infecting nodes. We have observed that the infection patterns are then less robust; in models where simple and complex contagion events can co-exist, the robustness of patterns and their similarity to the case of simple processes depends on the ratio between events of simple and complex contagions. In a threshold-based model, patterns differ more across parameter values. Fourth,

the similarity between the averaged infection patterns discussed here remains in all cases rather high, even between contagion processes of different nature. Both these results concerning complex contagion spreading patterns constitute a major new contribution to the literature, as we are aware of almost no result on this topic. Notably, the observed high similarity might at first glance seem to contradict a previous contribution, which showed that observing the propagation patterns of single processes makes it possible to distinguish between processes based on simple contagion, higher-order contagion, or threshold processes [14]. However, we consider here averages, which are indeed all correlated with the matrix of link weights describing the network, while [14] considered individual single realizations; moreover, the fact that spreading patterns are similar does not mean that they are indistinguishable, and indeed the results of [14] relied on machine learning techniques trained on a well-chosen set of features to manage to perform the distinction between different types of processes.

Our results have interesting implications that can impact our way of thinking about and performing numerical simulations of spreading processes for decision-making purposes. First, the extreme robustness of the spreading patterns for models of simple contagion implies that simulations of very schematic models with arbitrary parameters carry an enormous amount of information on the dynamics of spreading processes with apparently much more complicated dynamics. It is also possible to use these schematic models to provide a ranking of the risk of nodes to be reached, or of their spreading power: this ranking will indeed remain remarkably accurate for different processes. This is very important as, when a new disease emerges, it is initially difficult to estimate its parameters and sometimes even the types of compartments that should be taken into account in its modeling. Even in such cases, simulations with simplified models can thus bring interesting initial insights.

Second, even if single instances of simple and complex contagion processes present differences [14], it is also noteworthy that, when considering average infection patterns, their similarity remains high. Schematic simple contagion models can thus still be used to obtain information on the patterns of a social contagion process, and on the ranking of hosts in terms of their probability to be reached or their ability to propagate. However, the uncertainty on such ranking is higher than with simple contagion processes if the precise mechanism determining the propagation (e.g., depending on a threshold, or implying group effects) and the corresponding parameters are unknown.

Third, the stronger dependency of complex contagion processes on models and parameters implies the need for additional tools to determine whether an observed contagion process is determined by simple or complex contagion mechanisms. A first step in this direction was performed in [14], but more investigations, especially on real (social) contagion data, are desirable. Moreover, as the infection patterns depend on the ratio of contagion events occurring in pairwise events or in larger group, data collection efforts should explicitly target the measure of group interactions and not be restricted to pairwise representations of the system under scrutiny, in order to correctly inform models.

Our work has limitations worth mentioning, which also open some avenues for future work. The set of networks on which we have performed our investigation corresponds to diverse contexts of empirical contacts and thus entails a variety of complex interaction patterns, but remains limited. It would be interesting to extend our study to synthetic (hyper)networks where the distributions of degrees and of group sizes and the overlap between dyads and triads could be controlled. Our work also deals with static networks, and could be extended to temporal networks, especially as the propagation paths and infection risk might then be measured during a certain period while the propagation could then take place at another time [46, 47]. Finally, the infection patterns could also be studied for other models of complex contagion (including contagion events in groups of arbitrary sizes [48]).

## Methods

### Models of simple contagion

We consider three different epidemic processes, all of them agent-based compartmental models, i.e., in which each agent (represented by a node of the network) can pass through a finite set of possible compartments describing the evolution of a disease.

In the SIR model, a susceptible node $i$ (in compartment S) can become infected (changing compartment to I) by contact with one of its neighbors on the network $j$. This transition takes place with rate $\beta W_{ij}$, where $\beta$ is the infection rate, a free parameter of the model, and $W_{ij}$ is the weight of the connection between $i$ and $j$. Each node will then recover (becoming R) independently at rate $\mu_I$, another free parameter. We note that, as we consider processes occurring on static networks, rescaling all parameters by the same factor does not change the dynamics but only sets a global time scale. We thus consider for simplicity parameters of order 1 in all cases.

The SEIR model is similar to the previous one with the addition of one state: exposed (E). Newly infected individuals first enter the exposed (non-infectious) state and, with a rate $\mu_E$, they transition to the infectious state. Again, they will recover at rate $\mu_I$. We consider three versions of SEIR models: SEIRe1, SEIRe4, and SEIRi4, which only differ by the values of their parameters, which are given in Table 1. SEIRe1 is a baseline in which all rates are equal, and in each variation we change one of the parameters by a factor 4, making the average duration of the corresponding state four times longer (for instance in SEIRe4, a node spends on average four times more time in the exposed state than in the SEIRe1 version), so that these average durations differ significantly in the different models, but do not change order of magnitude (which would be unrealistic).

In both SIR and SEIR, the recovery rate $\mu_I$ and the exposed-to-infected rate $\mu_E$ are constant, implying that the times spent by an agent in the infected and exposed states are random variables drawn from exponential distributions with respective averages $\tau_I = 1/\mu_I$ and $\tau_E = 1/\mu_E$ (which are thus gamma distributions with standard deviations $\sigma_X = \tau_X$ with $X = I, E$). Instead of constant rates, we can also consider times in the E and I states distributed according to

**Table 1. Simple contagion model parameters.**

| SIR model | | | | | | |
|---|---|---|---|---|---|---|
| | $\mu_I$ | | | | | |
| SIR | 0.25 | | | | | |

| Markovian SEIR models | | | | | | |
|---|---|---|---|---|---|---|
| | $\mu_E$ | $\eta_E$ | $\mu_I$ | $\eta_I$ | | |
| SEIRe1 | 1 | 1 | 1 | 1 | | |
| SEIRe4 | 0.25 | 1 | 1 | 1 | | |
| SEIRi4 | 1 | 1 | 0.25 | 1 | | |

| Non-Markovian SEIR models | | | | | | |
|---|---|---|---|---|---|---|
| | $\mu_E$ | $\eta_E$ | $\mu_I$ | $\eta_I$ | | |
| SEIRe1v025 | 1 | 0.25 | 1 | 0.25 | | |
| SEIRe4v025 | 0.25 | 0.25 | 1 | 0.25 | | |
| SEIRi4v025 | 1 | 0.25 | 0.25 | 0.25 | | |

| COVID model | | | | | | |
|---|---|---|---|---|---|---|
| | $\tau_E \pm \sigma_E$ | $\tau_p \pm \sigma_p$ | $\tau_I \pm \sigma_I$ | $p_c$ | $r_p$ | $r_{sc}$ |
| COVID | 4 ± 2.3 | 1.8 ± 1.8 | 5 ± 2.0 | 0.5 | 0.55 | 0.55 |

gamma distributions with averages $\tau_E = 1/\mu_E$ and $\tau_I = 1/\mu_I$ and standard deviations $\sigma_X = \eta_X \tau_X$ with $\eta \neq 1$, thus obtaining non-markovian models. We consider the extension of the three versions of the SEIR model (SEIRe1, SEIRe4, and SEIRi4) to this non-markovian framework, namely SEIRe1v025, SEIRe4v025, and SEIRi4v025. In these models, the average durations $\tau_I$ and $\tau_E$ are the same as in the Markovian versions, but the standard deviations are reduced by a factor 4 with respect to the Markovian cases: this yields clearly different distribution of the durations of the states with respect to the Markovian case, without going to extreme, unrealistic cases (see Table 1).

We also consider the COVID model describing the propagation of SARS-CoV2 used in [13, 33]. In this model, when a susceptible agent is contaminated it transitions to an exposed state followed by a pre-symptomatic infectious state, remaining in these states for times extracted from gamma distributions with respective averages $\tau_E$ and $\tau_p$, and standard deviations $\sigma_E$ and $\sigma_p$. Then individuals can either evolve into a sub-clinical infection or manifest a clinical infection, with respective probabilities $1 - p_c$ and $p_c$. The duration in the infectious state is extracted from a gamma distribution with average $\tau_I$ and standard deviation $\sigma_I$. An individual $i$ in the infected states (pre-symptomatic, sub-clinical or clinical) can transmit the disease to a susceptible individual $j$ when in contact with it with respective rates of transmission $r_p \beta W_{ij}$, $r_{sc} \beta W_{ij}$, and $\beta W_{ij}$. We use here the same parameter values as in [13, 33].

Table 1 shows the values for the different parameters used in these models. Moreover, in all cases, the parameter $\beta$ is tuned to obtain a desired specific value for the basic reproductive number $R_0$, as detailed in the next section.

## Reproductive number and calibration of the simple contagion models

The reproductive number, $R_0$, is defined as the average number of cases directly generated by one infected individual in a population where all the others are susceptible. In detail, each simulation begins with one random infected node $i$ and we count all the neighbors of $i$ that are directly infected by it until $i$ becomes recovered, obtaining a potentially different value in each stochastic simulation. Averaging over these values at fixed parameters yields $R_0$.

Specifically, we perform 1000 simulations for 20 values of $\beta$ to obtain the corresponding values of $R_0$ (ranging between 1 and 4) and thus a correspondence table between $\beta$ and $R_0$. For each desired value of $R_0$, it is then enough to interpolate between the values in the table to obtain the value of $\beta$ needed in the simulations.

## Data sets

We use high-resolution face-to-face empirical contacts data collected using wearable sensors in different settings. The data sets are publicly available on the website http://www.sociopatterns.org/datasets. Data sets are available as lists of contacts over time (with a temporal resolution of 20 s) between anonymized individuals. The considered data sets are:

- **Primary school**, which describes the contacts among 232 children and 10 teachers in a primary school in Lyon, France, during two days of school activity in 2009 [49]. The school is composed of 5 grades, each of them comprising 2 classes, for a total of 10 classes.

- **Workplace**, gathering the contacts among 214 individuals, measured in an office building in France during two weeks in 2015 [27].

- **Hospital**, which describes the interaction among 42 health care workers (HCWs) and 29 patients in a hospital ward in Lyon, France, gathered during three days in 2010 [50].

- **High school**, describing the contacts among 324 students of "classes préparatoires" in Marseille, France, during one week in 2013 [51].

- **Conference**, which describes the interactions of 405 participants to the 2009 SFHH conference in Nice, France [52].

## From the data to weighted graphs and hypergraphs

As explained in the previous section, the data sets we use describe temporally resolved interactions between individuals. Each data set is provided as a list of interactions between individuals. Each element of the list corresponds to a time in which two individuals were registered as in interaction. Each such interaction event is reported in the form "$t\ i\ j$" where $t$ indicates the time, with a temporal resolution of 20 seconds, and $i$ and $j$ the involved individuals identification numbers.

For each data set we obtain a weighted static network by aggregating over time as follows:

- each individual involved in the data collection is represented by a node of the network;

- each pair of nodes $(i,j)$ appearing in the list of events is represented as a link $ij$ between nodes $i$ and $j$ in the network;

- we denote by $n_{ij}$ the number of times that the pair $(i, j)$ appears in the data set (the total contact duration between the corresponding individuals is thus $n_{ij}$ times 20 seconds);

- we compute the maximum of these numbers over all pairs of individuals, $n_{max} = \max_{i,j} n_{ij}$;

- the weight of the link $ij$ is given by $n_{ij}/n_{max}$.

We then use the weighted networks resulting from this procedure to simulate the spreading models of simple and threshold contagion, in which contagion events involve only links.

We moreover use the data sets to build weighted hypergraphs involving both links (hyperedges of size 2, or "first-order interactions") and hyperedges of size 3 (so-called "second-order interactions", i.e., interactions between 3 nodes). We build the hypergraphs as in [21]. Namely, we first consider all the links of the weighted graphs obtained as above: these links form the weighted hyperedges of size 2 of the hypergraph. To build the second-order interactions, we first identify all the simultaneous interactions at each time $t$, obtaining so-called "snapshot graphs": the snapshot graph at time $t$ is simply the network of all interactions taking place at $t$. In each snapshot graph, we identify its cliques (sets of nodes all interacting with each other) of size at least 3. For instance, if at $t$ the interactions $(i, j)$, $(i, k)$, $(j, k)$ are present, then $ijk$ is a clique at time $t$. If a clique of size larger than 3 is present, such as $ijkl$, we decompose it into all the possible triads, here $ijk, ijl, ikl, jkl$. We then proceed as for the weighted graphs, namely

- each triad $ijk$ appearing at least in one snapshot becomes a hyperedge of size 3 of the weighted hypergraph;

- we denote by $n_{ijk}$ the number of snapshots in which the triad $ijk$ appears;

- we compute the maximum of these numbers over all triads of individuals, $n^{(2)}_{max} = \max_{i,j,k} n_{ijk}$;

- the weight of the hyperedge $ijk$ is given by $n_{ijk}/n^{(2)}_{max}$.

We use the resulting weighted hypergraphs in the numerical simulations of the simplicial contagion processes.

### Cosine similarity

The cosine similarity $cs(v, w)$ quantifies the similarity between two vectors $\mathbf{v}$ and $\mathbf{w}$ of the same dimension $n$. It is defined as:

$$cs(\mathbf{v}, \mathbf{w}) = \frac{\mathbf{v} \cdot \mathbf{w}}{\|\mathbf{v}\| \, \|\mathbf{w}\|} = \frac{\sum_{i=1}^{n} v_i w_i}{\sqrt{\sum_{i=1}^{n} v_i^2} \sqrt{\sum_{i=1}^{n} w_i^2}} \quad . \tag{1}$$

It is bounded in $[-1, 1]$. It is equal to 1 when one vector is proportional to the other with a positive proportionality factor, and to 0 if they are orthogonal.

In order to measure the similarity between two infection patterns we generate two vectors from the corresponding $C$ matrices (concatenating all the rows of one matrix) and we apply the definition of cosine similarity to the two resulting vectors. Since all the elements of $C$ are non-negative, the cosine similarity is here bounded in $[0, 1]$.

## Supporting information

**S1 Text. Supporting information.** Supporting information is provided in a separate pdf file. It contains additional analyses on infection pattern, attack rate, spreader and receiver index executed for the primary school dataset used in the main text and for additional datasets.
(PDF)

## Author Contributions

**Conceptualization:** Diego Andrés Contreras, Giulia Cencetti, Alain Barrat.

**Formal analysis:** Diego Andrés Contreras, Giulia Cencetti.

**Investigation:** Diego Andrés Contreras, Giulia Cencetti, Alain Barrat.

**Supervision:** Alain Barrat.

**Visualization:** Giulia Cencetti.

**Writing – original draft:** Diego Andrés Contreras, Giulia Cencetti, Alain Barrat.

**Writing – review & editing:** Giulia Cencetti, Alain Barrat.

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
