## [Decision Letter · Decision Letter 0]

16 Jan 2024

Dear Miss Cencetti,

Thank you very much for submitting your manuscript "Infection patterns in simple and complex contagion processes on networks" for consideration at PLOS Computational Biology.

As with all papers reviewed by the journal, your manuscript was reviewed by members of the editorial board and by several independent reviewers. In light of the reviews (below this email), we would like to invite the resubmission of a significantly-revised version that takes into account the reviewers' comments.

The manuscript is generally well received by the reviewers, although some issues were highlighted that would improve the clarity of the work. Particularly, for readers to assess the used approach, decisions in the methods section need more explanation and elaboration, Further, please consider a broader elaboration on the implications of the findings for the broader modeling community.

We cannot make any decision about publication until we have seen the revised manuscript and your response to the reviewers' comments. Your revised manuscript is also likely to be sent to reviewers for further evaluation.

Sincerely,

Quirine ten Bosch

Academic Editor

PLOS Computational Biology

Thomas Leitner

Section Editor

PLOS Computational Biology

The manuscript is generally well received by the reviewers, although some issues were highlighted to good improve the clarity of the work. Particularly, for readers to assess the approach, decisions in the methods section need more explanation/elaboration, Further, please consider a broader elaboration on the implications of the findings for the broader modeling community.

Reviewer's Responses to Questions

**Comments to the Authors:**

Reviewer #1: Contreras, Cencetti, and Barrat have submitted a manuscript entitled “Infection patterns in simple and complex contagion processes on networks” for publication in PLOS Computational Biology. Overall, this is a great numerical simulation paper in computational epidemiology. Modulo some notes below, publication in PLOS Computational Biology is clearly warranted. When I consider the contributions of the paper, I see the following primary points:

(a) A clear formulation of problem, with numerical insights closely associated with (and thoroughly explored through) the model parameters

(b) The use of empirical datasets—and, when non-trivial patterns do occur, the explanations with regards to the particular features of the datasets.

(c) A simple-to-understand story line: Simple infection patterns are largely similar, but group contagions depend on an interplay between pairwise and higher-order parameters. The threshold contagion may be made similar to the simple one, but only when the threshold parameter is large.

Notably not on the list are:

(1) The abstract is on the lengthy end. It would be nice if the authors could trim it to 1/2 to 2/3 of the current size.

(2) In Section II-C, and regarding (b) above, it would be nice if the authors could supplement a contrasting numerical experiment, based on the same scaffold (i.e., same edges) of an empirical social network, but artificially redesign the weights in some way, such that the higher-order interactions and the pairwise ones do not necessarily overlap, and re-run the experiments at the end. This is to test whether the explanation—that both simple and higher-order mechanisms tend to use the same infection routes, when the weights are correlated—is probable or not.

(3) Adding to (2), this kind of dataset could be a network where participants spend time with families and friends, but also a significant time with colleagues. With colleagues (who are not friends or family members), the W_{ijk} is large, but the pairwise W_{ij}, W_{jk}, W_{ik} are small. I am not sure if sociopatterns contain this type of dataset.

(4) In the Lines between 354-362 (in Section II-C), is there a reason that the authors choose ½ and ½ for the contributing C_{ij} weights. Why not choose numbers (which sums to 1) with proportion to W_{ij} and W{lj}?

(5) In the last paragraph of Section II-D, the sentences are not very comprehensible to me. For example, what does the *it* in Line 475 refer to? (I confess that I jumped to reading this paragraph, and noticed that I did not understand its meaning.)

(6) Please note that all the points (1)-(5) are minor.

Reviewer #2: To the authors of the manuscript titled “Infection patterns in simple and complex contagion processes on networks”,

In this manuscript, the authors aim to understand the differences in infection patterns for given network structures using different contagion models. The manuscript is interesting and well-written, however, there are important points that need clarification or elaboration:

1. The Discussion section of the study requires significant elaboration on the implications of the results and findings. The authors are expected to address how other researchers, decision makers, and/or policy makers can use and derive benefits from these findings. Furthermore, the authors can consider mentioning the broader implications for the infectious diseases research community.

2. The contribution of the findings of this study to the existing knowledge base needs elaboration.

3. The Methods section, in general, is expected to provide more information for understanding the findings and reproducing the results presented:

3.1. The set of equations, pseudocodes or visualizations of the simulation models would enhance comprehension of the model structures. These models may be intuitive to many researchers but require elaboration given the broader audience of PLOS CB.

3.2. The selection of the Model Parameters in Table 1 needs explanation.

3.3 How the parameter beta is tuned to obtain a desired specific value for the basic reproduction number R0 needs elaboration.

3.4. The calibration process under the heading “B. Reproductive number and calibration” requires clarification.

3.5. Among the 5 datasets (primary school, workplace, hospital, high school and conference), the rationale behind selecting the primary school dataset for the main body of the manuscript needs elaboration.

3.6. The calculations for the weights of the links between nodes need elaboration. What are the assumptions behind these calculations? Could you provide the steps required to create the same weighted graphs using the dataset references you provided?

3.7. As far as I understand, the authors use weighted graphs for “simple models”, while they use weighted hypergraphs for “simplicial models” and “threshold models”. If this is the case, it is expected to be explicitly stated in Methods.

3.8. Again, as far as I understand, weighted graphs are formed by using the average daily time between contacts, while weighted hypergraphs are formed by using the number of times that the connection has appeared in the data. The authors are expected to explain the reasoning behind this difference, and any observations they find resulting from this difference.

3.9. The general idea/formula for cosine similarity is required.

3.10. Last but not least, the platform on which these simulations and analyses were conducted is expected to be stated. The simulation models should be able to be run/repeatable by other researchers.

4. Page 3, lines 275-277: “This suggests that the infection pattern of a spreading model mostly depends on its average final attack rate.”  I believe that this statement may be overly strong, since this is an observation for a given range of attack rates. For instance, what percentage of the simulations fall into the range 0.75 < a < 0.85? Do the authors observe the same phenomena for other ranges of attack rates as well? (This may bring us to the following point). I believe the generalizability of this statement needs to be reconsidered.

5. Page 4, lines 307 – 310: “In other words, at each time step the infection pattern (which describes the contagion probability of each connection until that time) is almost completely determined by the attack rate reached at that specific time.” � I, unfortunately, could not fully comprehend that conclusion. Some clarification on the construction of Figure 3 may help other readers to understand that statement. Do results in Figure 3 correspond to the simulation of 1000 runs, or less, or just 1 example run for each panel and each R0? How were the distributions of attack rates at the bottom of Figure 3 formed? Were the distributions formed using multiple runs? In general, the chain of thought from the observations in Figure 3 to the conclusion stated in lines 307-310 needs further clarification.

6. A visualization demonstrating how simplicial contagion models look would be helpful for the readers.

**Have the authors made all data and (if applicable) computational code underlying the findings in their manuscript fully available?**

Reviewer #1: **No: **Data are available, but not the computational code.

Reviewer #2: **No: **

PLOS authors have the option to publish the peer review history of their article (what does this mean?). If published, this will include your full peer review and any attached files.

Reviewer #1: No

Reviewer #2: No
---

## [Decision Letter · Decision Letter 1]

9 Apr 2024

Dear ` Cencetti,

Thank you very much for submitting your manuscript "Infection patterns in simple and complex contagion processes on networks" for consideration at PLOS Computational Biology. As with all papers reviewed by the journal, your manuscript was reviewed by members of the editorial board and by several independent reviewers. The reviewers appreciated the attention to an important topic. Based on the reviews, we are likely to accept this manuscript for publication, providing that you modify the manuscript according to the review recommendations.

Sincerely,

Quirine ten Bosch

Academic Editor

PLOS Computational Biology

Thomas Leitner

Section Editor

PLOS Computational Biology

The reviewers and editors appreciate the improvements made to the manuscript. Some minor comments remain regarding the discussion and the code base that we would like to see addressed.

Reviewer's Responses to Questions

**Comments to the Authors:**

Reviewer #1: Contreras, Cencetti, and Barrat have resubmitted their manuscript titled “Infection patterns in simple and complex contagion processes on networks” for publication in PLOS Computational Biology.

The manuscript is now in good standing for publication. In particular, the authors have properly addressed the two reviewers' previous comments: the newly added Figure 1, comments, experiments, and the re-written paragraphs all build a stronger connection to the PLOS CB readership.

Reviewer #2: To the authors of "Infection patterns in simple and complex contagion processes on networks",

I thank the authors for addressing the comments so carefully and thoroughly. I believe the revision of the manuscript has significantly improved its comprehensibility, making it more accessible to a wider audience. I have some additional comments about the manuscript:

- Even though the authors showed significant effort in editing the Discussion section, I think the comment saying "The contribution of the findings of this study to the existing knowledge base needs elaboration." has not been addressed yet. The authors are suggested to elaborate on the interplay between their findings and the available literature and summarize how their findings align with or oppose the current knowledge. In the current version of the Discussion section, the relationship between the manuscript's implications and the existing studies has not been presented, which makes it harder for the readers to comprehend the concrete contribution of the work done by the authors.

- The code for the models are available on Github but the models cannot be run (at least by me), so the results on the manuscript are not reproducible. While trying to run the models, I got errors because of missing files or functions. The authors are suggested to enrich the ReadMe fie on the repository with a summary of instructions on how to reproduce the findings presented in the manuscript.

- Minor comment: In Section S7, the references for the datasets are broken and shown with [?].

**Have the authors made all data and (if applicable) computational code underlying the findings in their manuscript fully available?**

Reviewer #1: Yes

Reviewer #2: **No: **The code for the models are available on Github but the models cannot be run (at least by me), so the results on the manuscript are not reproducible. While trying to run the models, I got errors because of missing files or functions. The authors are suggested to enrich the ReadMe fie on the repository with a summary of instructions on how to reproduce the findings presented in the manuscript.

PLOS authors have the option to publish the peer review history of their article (what does this mean?). If published, this will include your full peer review and any attached files.

Reviewer #1: No

Reviewer #2: No

Figure Files:

Data Requirements:

Reproducibility:

References:

---

## [Editor Report · Decision Letter 2]

28 May 2024

Dear ` Cencetti,

We are pleased to inform you that your manuscript 'Infection patterns in simple and complex contagion processes on networks' has been provisionally accepted for publication in PLOS Computational Biology.

Best regards,

Quirine ten Bosch

Academic Editor

PLOS Computational Biology

Thomas Leitner

Section Editor

PLOS Computational Biology

In the final iteration of the manuscript, the authors have addressed the final comments from reviewer 2, by more explicitly relating the results to existing literature. The authors have further updated to code base to ensure reproducibility of the results. With those final iterations, the manuscript is now suitable for publication in PLOS Computational Biology.

---

## [Editor Report · Acceptance letter]

5 Jun 2024

PCOMPBIOL-D-23-01486R2 

Infection patterns in simple and complex contagion processes on networks

Dear Dr Cencetti,

I am pleased to inform you that your manuscript has been formally accepted for publication in PLOS Computational Biology. Your manuscript is now with our production department and you will be notified of the publication date in due course.

With kind regards,

Anita Estes
